# Association between long COVID and nonsteroidal anti-inflammatory drug use by patients with acute-phase COVID-19: A nationwide Korea National Health Insurance Service cohort study

Ye-Seul Lee[1], Heejun Kim[1], Sunoh Kwon[2]*, Tae-Hun Kim[3]*

1 Jaseng Spine and Joint Research Institute, Jaseng Medical Foundation, Seoul, Republic of Korea,
2 Korean Medicine Convergence Research Division, Korea Institute of Oriental Medicine, Daejeon, Republic of Korea, 3 Korean Medicine Clinical Trial Center, Korean Medicine Hospital, Kyung Hee University, Seoul, Republic of Korea

☯ These authors contributed equally to this work.
* rockandmineral@gmail.com (THK); sunohkwon@kiom.re.kr (SK)

**Data Availability Statement:** All data used in this study is suggested within the paper itself. The metadata related to the analysis of this study can

## Abstract

### Introduction

Long coronavirus disease (COVID) poses a significant burden following the coronavirus disease 2019 (COVID-19) pandemic. Debate persists regarding the impact of nonsteroidal anti-inflammatory drug (NSAID) administration during acute-phase COVID-19 on the development of long COVID. Hence, this study aimed to assess the potential association between NSAID use and long COVID using data from patients with COVID-19 in Korea's National Health Insurance Service.

### Methods

This nested case-control study defined the study cohort as patients diagnosed with COVID-19 for the first time between 2020 and 2021. The primary exposure investigated was NSAID prescriptions within 14 days of the initial COVID-19 diagnosis. We used propensity score matching to create three control patients matched to each patient in the NSAID exposure group. Odds ratios (ORs) and 95% confidence intervals (CIs) were calculated after the adjustment for demographics, Charlson Comorbidity Index, and existing comorbidities.

### Results

Among the 225,458 patients diagnosed with COVID-19, we analyzed data from 254 with long COVID. The adjusted OR (aOR) for NSAID exposure during acute-phase COVID-19 was higher in long COVID cases versus controls (aOR, 1.79; 95% CI, 1.00–3.19), suggesting a potential relationship. However, a sensitivity analysis revealed that the increased odds of NSAID exposure in the acute phase became statistically non-significant (aOR, 1.64; 95% CI, 0.90–2.99) when COVID-19 self-quarantine duration was included as a covariate. Additionally, acetaminophen exposure was not significantly associated (aOR, 1.12; 95% CI,

be found in the supporting information. Further materials for future additional analysis are available upon request and review by the National Health Insurance Service of Korea, and can only be accessed at a designated location for a limited time period for protection of personal information. If data for further analysis is necessary, the application form for data analysis can be submitted online according to the Data Request Guide from the National Health Insurance Sharing Service. After the application is reviewed by the National Health Insurance Service, access to the data will be granted, and the actual data can be accessed. For details regarding application and submission, check the following webpage: https://nhiss.nhis.or.kr/en/a/a/150/lpaa150m01en.do.

**Funding:** This study was supported by the Korea Institute of Oriental Medicine [grant number KSN2121220]. The funders had no role in study design, data collection and analysis, decision to publish, or preparation of the manuscript.

**Competing interests:** The authors have declared that no competing interests exist.

**Abbreviations:** APAP, acetaminophen or paracetamol; ACE2, angiotensin-converting enzyme 2; CCI, Charlson Comorbidity Index; CIs, confidence interval; ICU, intensive care unit; ECMO, extracorporeal membrane oxygenation; ICD-10, International Classification of Diseases 10th Revision; NHIS, National Health Insurance Service; NSAIDs, nonsteroidal anti-inflammatory drugs; ORs, odds ratio; PS, propensity score; SARS-CoV-2, severe acute respiratory syndrome coronavirus 2; SMD, standard mean difference.

0.75–1.68), while antiviral drugs demonstrated a stronger association (aOR, 3.75; 95% CI, 1.66–8.48).

## Conclusion

Although this study suggests a possible link between NSAID use in the acute COVID-19 infection stage and a higher risk of long COVID as well as both NSAID and acetaminophen use during the chronic COVID-19 period and a lower risk of long COVID, the association was not statistically significant. Further research is needed to determine the causal relationship between the various treatment options for acute COVID-19 and the development of long COVID.

## Introduction

As the number of new coronavirus disease 2019 (COVID-19) cases has significantly decreased, the focus on patient care has shifted from treating acute-phase patients to managing patients with post-acute or chronic symptoms. The term "long COVID," as defined by the European Society of Clinical Microbiology and Infectious Diseases, refers to symptoms persisting beyond 12 weeks after a COVID-19 diagnosis with no identifiable causes [1]. The prevalence of long COVID varies depending on the definition of the disease and duration of patient follow-up. For instance, the UK Office for National Statistics reported that 22% of individuals remain symptomatic at 5 weeks post-infection, a figure that decreases to 10% at 12 weeks [2]. In Korea, a survey suggested that, while approximately 50% of patients with COVID-19 experience related symptoms, only 5% receive outpatient treatment [3]. Common long COVID symptoms include fatigue, anxiety, cognitive impairment, and insomnia. Notably, 40% of respondents reported feeling depression or anxiety, highlighting the significant burden of long COVID [3].

The use of nonsteroidal anti-inflammatory drugs (NSAIDs) during the acute phase of COVID-19 remains controversial. Evidence suggests that NSAID use may worsen acute symptoms by increasing the expression of angiotensin-converting enzyme 2 (ACE2), a receptor used by the virus, allowing severe acute respiratory syndrome coronavirus 2 (SARS-CoV-2) to enter the cells [4]. A Korean study analyzing the data of 1,824 hospitalized COVID-19 patients found that those who used NSAIDs had a 54% higher risk of a severe outcome (death, intensive care unit [ICU] admission, ventilator use, or sepsis) than non-users [5].

The exact causes of long COVID remain unclear, but long-term organ damage due to the virus's potential for systemic cellular penetration is suspected [2]. Investigating the possible mechanisms of long COVID is crucial, and examining the role of NSAID use during the acute phase may offer valuable insights.

This study utilized data of patients with COVID-19 in the Korean National Health Insurance Service (NHIS) to explore the potential association between NSAID use during acute-phase COVID-19 and the development of long COVID with the aim of shedding light on the potential mechanisms.

## Materials and methods

### Data source

This study was exempted from review by the Institutional Review Board of Kyung Hee University (KHSIRB-22-563(EA)). We employed a nationwide, retrospective, nested case-control

study design that utilized healthcare claims data retrieved from the Korean NHIS. This database encompasses nearly the entire South Korean population (97% of citizens are subscribed [6], while an additional 3.0% are covered by the Medical Aid program [7]), making it nationally representative. Recorded information obtained from Korean Statistics included demographics, inpatient and outpatient medical services, diagnostic codes, drug prescriptions, healthcare providers, and dates of death. Related materials and metadata are available on the National Health Insurance Data Sharing Service website (http://nhiss.nhis.or.kr [accessed June 5, 2024]).

## Study cohort

This nested case-control study included patients with a first-time COVID-19 diagnosis (using the International Classification of Diseases 10th Revision [ICD-10] codes specific to COVID-19: U07.1, U07.2, B34.2, and B97.2) between October 1, 2020, and December 31, 2021. The date of the initial diagnosis served as the entry date. Consecutive COVID-19 records within 10 days were considered a single episode to account for Korea's self-quarantine policy in 2020–2021. The number and date of each diagnosis were documented for patients with multiple episodes.

Patients with long COVID, defined as symptoms persisting for at least 90 days after the initial COVID-19 diagnosis [2, 8], were identified using ICD-10 code U09 (post-COVID-19 condition, unspecified) within the defined COVID-19 cohort. To ensure that this criterion was met, we analyzed the most recent COVID-19 diagnosis date before a long COVID diagnosis and excluded patients diagnosed with long COVID at <90 days after their most recent COVID-19 episode. Additionally, patients with less than a 90-day pre-diagnosis period within the data source were excluded because of limited covariate information.

Controls were defined as patients without diagnosed long COVID throughout the study period with a minimum follow-up period of 90 days after their COVID-19 diagnosis. Patients with repeated COVID-19 diagnoses within the 90-day window were excluded (Fig 1).

## Exposure

The main exposures of interest were the prescription of various NSAIDs, including aceclofenac, diclofenac, etodolac, dexibuprofen, ibuprofen, ketoprofen, dexketoprofen, ketorolac, tromethamine, meloxicam, naproxen, nimesulide, piroxicam, celecoxib, polmacoxib, etoricoxib, mefenamic acid, and tiaprofenic acid. The Anatomical Therapeutic Chemical Classification System codes for each medication are listed in **Table 1**. We focused on the acute phase after a COVID-19 diagnosis and defined the exposure window as 14 days after the initial diagnosis. For the sensitivity analysis, we extended the exposure window to encompass the entire follow-up period, which was at least 90 days for both cases and controls.

## Covariates

We assessed various medical conditions that potentially influenced the development of long COVID or the use of medications. These conditions included prescriptions for acetaminophen or paracetamol (APAP), aspirin, oral steroids, and antiviral medications used for COVID-19 in 2020 and 2021. The antiviral medications included ritonavir, oseltamivir phosphate, and ribavirin. Demographic covariates included sex, age, and geographic area of COVID-19 diagnosis. The year and month of diagnosis were included as string variables. The Charlson Comorbidity Index (CCI) was calculated using claims data from 90 days before the COVID-19 diagnosis with consideration of various health conditions such as diabetes mellitus, chronic back pain, osteoarthritis, rheumatoid arthritis, osteoporosis, chronic obstructive pulmonary

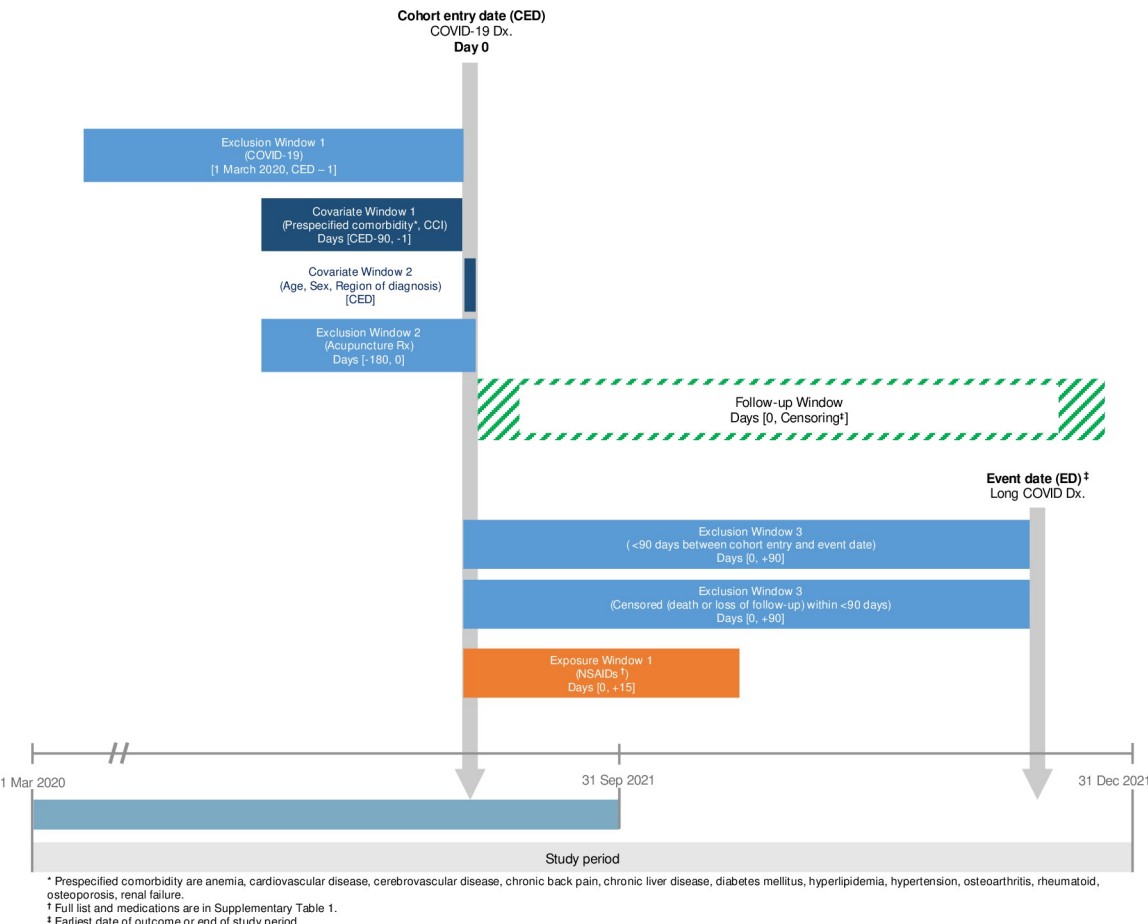

**Fig 1. Study design.**

disease, dementia, schizophrenia, depressive disorder, anxiety disorders, hyperlipidemia, hypertension, cardiovascular diseases, renal failure, chronic liver diseases, cerebrovascular diseases, anemia, sleep disorders, and thyroid disorders. Specific diseases and their codes are listed in **S1 Table**.

## Statistical methods

Descriptive statistics, including demographic variables, comorbidities, CCI scores, and medication use, were used to summarize the participants' baseline characteristics. The effect size for continuous variables was assessed using standardized mean differences, where an absolute value $\geq 0.2$ indicated a relevant intergroup difference [9]. To account for potential baseline differences between cases and controls, propensity score (PS) matching was employed. The PS were derived from the predicted probability of developing long COVID with consideration of factors such as the year and month of the COVID-19 diagnosis, region of diagnosis, and demographic factors such as sex, age group, and CCI. We used exact matching for core characteristics (year and month, age group, sex, and CCI score) and the greedy nearest-neighbor method with a caliper of 0.1 for the PS matching. This method aims to identify controls with similar predicted probabilities for a long COVID period for each case.

**Table 1.  Patient characteristics before versus after propensity score matching.**

| Variable | Before | | | After | | |
|---|---|---|---|---|---|---|
| | Long COVID (n = 254) | Control (n = 225,204) | SMD | Long COVID (n = 254) | Control (n = 755) | SMD |
| **Sex, n (%)** | | | | | | |
| Male | 123 (48.4) | 116,152 (51.6) | -0.06 | 123 (48.4) | 365 (48.3) | 0.00 |
| Female | 131 (51.6) | 109,052 (48.4) | 0.06 | 131 (51.6) | 390 (51.7) | 0.00 |
| **Age, years, n (%)** | | | | | | |
| <30 | 24 (9.5) | 73,958 (32.8) | -0.60 | 24 (9.5) | 72 (9.5) | 0.00 |
| 30–39 | 31 (12.2) | 33,786 (15.0) | -0.08 | 31 (12.2) | 93 (12.3) | 0.00 |
| 40–49 | 32 (12.6) | 36,677 (16.3) | -0.11 | 32 (12.6) | 94 (12.5) | 0.00 |
| 50–59 | 56 (22.1) | 38,069 (16.9) | 0.13 | 56 (22.1) | 168 (22.3) | 0.00 |
| 60–69 | 62 (24.4) | 27,181 (12.1) | 0.32 | 62 (24.4) | 186 (24.6) | -0.01 |
| 70–79 | 35 (13.8) | 10,089 (4.5) | 0.33 | 35 (13.8) | 102 (13.5) | 0.01 |
| ≥ 80 | 14 (5.5) | 5,444 (2.4) | 0.16 | 14 (5.5) | 40 (5.3) | 0.01 |
| **CCI, n (%)** | | | | | | |
| 0 | 137 (53.9) | 171,177 (76.0) | -0.48 | 137 (53.9) | 408 (54.0) | 0.00 |
| 1 | 63 (24.8) | 32,040 (14.2) | 0.27 | 63 (24.8) | 187 (24.8) | 0.00 |
| 2 | 25 (9.8) | 12,623 (5.6) | 0.16 | 25 (9.8) | 75 (9.9) | 0.00 |
| ≥3 | 29 (11.4) | 9,364 (4.2) | 0.27 | 29 (11.4) | 85 (11.3) | 0.01 |
| **Area, n (%)** | | | | | | |
| Seoul | 112 (44.1) | 71,122 (31.6) | 0.26 | 112 (44.1) | 332 (44.0) | 0.00 |
| Metropolitan city | 27 (10.6) | 39,235 (17.4) | -0.20 | 27 (10.6) | 80 (10.6) | 0.00 |
| Province | 112 (44.1) | 113,162 (50.3) | -0.12 | 112 (44.1) | 336 (44.5) | -0.01 |
| Quarantine station | 3 (1.2) | 1,685 (0.8) | 0.04 | 3 (1.2) | 7 (0.9) | 0.02 |
| **Comorbidities, n (%)** | | | | | | |
| Diabetes | 51 (20.1) | 18,633 (8.3) | 0.34 | 51 (20.1) | 140 (18.5) | 0.04 |
| Chronic back pain | 35 (13.8) | 23,982 (10.7) | 0.10 | 35 (13.8) | 98 (13.0) | 0.02 |
| Osteoarthritis | 32 (12.6) | 14,619 (6.5) | 0.21 | 32 (12.6) | 87 (11.5) | 0.03 |
| Rheumatoid | 5 (2.0) | 1,596 (0.7) | 0.11 | 5 (2.0) | 13 (1.7) | 0.02 |
| Osteoporosis | 10 (3.9) | 5,375 (2.4) | 0.09 | 10 (3.9) | 28 (3.7) | 0.01 |
| Chronic obstructive pulmonary disease | 3 (1.2) | 789 (0.4) | 0.10 | 3 (1.2) | 4 (0.5) | 0.07 |
| Dementia | 5 (2.0) | 2,959 (1.3) | 0.05 | 5 (2.0) | 15 (2.0) | 0.00 |
| Schizophrenia | - | 1,080 (0.5) | -0.10 | - | 5 (0.7) | -0.12 |
| Depressive disorder | 5 (2.0) | 6,216 (2.8) | -0.05 | 5 (2.0) | 33 (4.4) | -0.14 |
| Anxiety disorder | 10 (3.9) | 6,686 (3.0) | 0.05 | 10 (3.9) | 28 (3.7) | 0.01 |
| Hyperlipidemia | 81 (31.9) | 34,697 (15.4) | 0.40 | 81 (31.9) | 229 (30.3) | 0.03 |
| Hypertension | 71 (28.0) | 32,466 (14.4) | 0.34 | 71 (28.0) | 193 (25.6) | 0.05 |
| Cardiovascular disease | 22 (8.7) | 7,051 (3.1) | 0.24 | 22 (8.7) | 39 (5.2) | 0.14 |
| Renal failure | 5 (2.0) | 1,305 (0.6) | 0.12 | 5 (2.0) | 14 (1.9) | 0.01 |
| Chronic liver disease | 37 (14.6) | 12,641 (5.6) | 0.30 | 37 (14.6) | 80 (10.6) | 0.12 |
| Cerebrovascular disease | 13 (5.1) | 4,363 (1.9) | 0.17 | 13 (5.1) | 37 (4.9) | 0.01 |
| Anemia | 19 (7.5) | 4,505 (2.0) | 0.26 | 19 (7.5) | 23 (3.1) | 0.20 |
| Sleep disorder | 22 (8.7) | 7,421 (3.3) | 0.23 | 22 (8.7) | 33 (4.4) | 0.17 |
| Thyroid disorder | 20 (7.9) | 8,439 (3.8) | 0.18 | 20 (7.9) | 33 (4.4) | 0.15 |
| **COVID diagnosis date, n (%)** | | | | | | |
| 2020.03. | - | 5 (0.0) | -0.01 | - | - | - |
| 2020.04. | - | 100 (0.0) | -0.03 | - | - | - |
| 2020.05. | - | 107 (0.1) | -0.03 | - | - | - |

(*Continued*)

**Table 1.**  (Continued)

| Variable | Before | | | After | | |
|---|---|---|---|---|---|---|
| | Long COVID (n = 254) | Control (n = 225,204) | SMD | Long COVID (n = 254) | Control (n = 755) | SMD |
| 2020.06. | - | 108 (0.1) | -0.03 | - | - | - |
| 2020.07. | - | 76 (0.0) | -0.02 | - | - | - |
| 2020.08. | - | 102 (0.1) | -0.03 | - | - | - |
| 2020.09. | - | 76 (0.0) | -0.02 | - | - | - |
| 2020.10. | 4 (1.6) | 1,678 (0.8) | 0.08 | 4 (1.6) | 9 (1.2) | 0.03 |
| 2020.11. | 18 (7.1) | 6,353 (2.8) | 0.20 | 18 (7.1) | 53 (7.0) | 0.00 |
| 2020.12. | 32 (12.6) | 20,407 (9.1) | 0.11 | 32 (12.6) | 96 (12.7) | 0.00 |
| 2021.01. | 25 (9.8) | 13,706 (6.1) | 0.14 | 25 (9.8) | 75 (9.9) | 0.00 |
| 2021.02. | 18 (7.1) | 8,942 (4.0) | 0.14 | 18 (7.1) | 54 (7.2) | 0.00 |
| 2021.03. | 16 (6.3) | 10,676 (4.7) | 0.07 | 16 (6.3) | 48 (6.4) | 0.00 |
| 2021.04. | 19 (7.5) | 15,539 (6.9) | 0.02 | 19 (7.5) | 57 (7.6) | 0.00 |
| 2021.05. | 29 (11.4) | 14,384 (6.4) | 0.18 | 29 (11.4) | 85 (11.3) | 0.01 |
| 2021.06. | 19 (7.5) | 13,169 (5.9) | 0.07 | 19 (7.5) | 56 (7.4) | 0.00 |
| 2021.07. | 39 (15.4) | 33,060 (14.7) | 0.02 | 39 (15.4) | 117 (15.5) | 0.00 |
| 2021.08. | 24 (9.5) | 42,060 (18.7) | -0.27 | 24 (9.5) | 72 (9.5) | 0.00 |
| 2021.09. | 11 (4.3) | 42,136 (18.7) | -0.46 | 11 (4.3) | 33 (4.4) | 0.00 |
| 2021.10. | - | 2,520 (1.1) | -0.15 | - | - | - |

CCI, Charlson comorbidity index; COVID, coronavirus disease; SMD, standard mean difference

The primary analysis assessed the potential relationship between a long COVID diagnosis and NSAID use during the acute COVID-19 period (0–14 days post-diagnosis). We compared NSAID exposure rates between long COVID cases and controls within the PS-matched and original cohorts. Odds ratios (ORs) and 95% confidence intervals (CIs) were calculated using univariate and multivariate logistic models after the adjustment for demographic factors, CCI, and comorbidities.

Four additional analyses were performed to explore the relationship between medication use and outcomes using the univariate and multivariate logistic regression models. Each analysis estimated ORs with 95% confidence intervals (CIs). We investigated the association between NSAID use during the chronic period (15 days post-diagnosis to the end of follow-up) and the development of long COVID. The follow-up ended with a long COVID diagnosis, a subsequent COVID-19 diagnosis, or conclusion of the observation period. Second, this analysis examined the association between NSAID use throughout the entire follow-up period and long COVID. Third and fourth, we replaced the outcome variable, long COVID, with APAP and antiviral drug use to assess the relationship with long COVID using the same models as before.

We conducted two sensitivity analyses to verify the robustness of the primary findings. First, we included the duration of COVID-19 episodes or self-quarantine as covariates in the analyses examining NSAID, APAP, and antiviral drug use during the post-COVID-19 period. Second, we limited the number of cases and controls to those who used NSAIDs or APAP alone. This allowed us to more directly compare the medication exposure rates between long COVID cases and controls. All statistical analyses were performed using SAS Enterprise Guide 7.1.

## Results

### Patient characteristics

Of the 225,458 patients diagnosed with COVID-19 in South Korea between October 1, 2020, and December 31, 2021, only 0.01% (254 patients) were subsequently diagnosed with long COVID (Fig 2). Imbalances were observed in the baseline variables between patients with long COVID and controls. Specifically, patients with long COVID were characterized by older age (50–79 years) and higher CCI scores. These patients also exhibit a higher prevalence of several comorbidities, including diabetes mellitus, osteoarthritis, hyperlipidemia, hypertension, cardiovascular diseases, chronic liver diseases, anemia, and sleep disorders. After the application of PS matching to create comparable groups, balance was achieved for demographic factors such as sex and age group, CCI, and underlying comorbid conditions except for anemia, which showed a higher prevalence of comorbidities than controls (standard mean difference [SMD] = 0.20), and the year and month of the COVID-19 episode (Table 1).

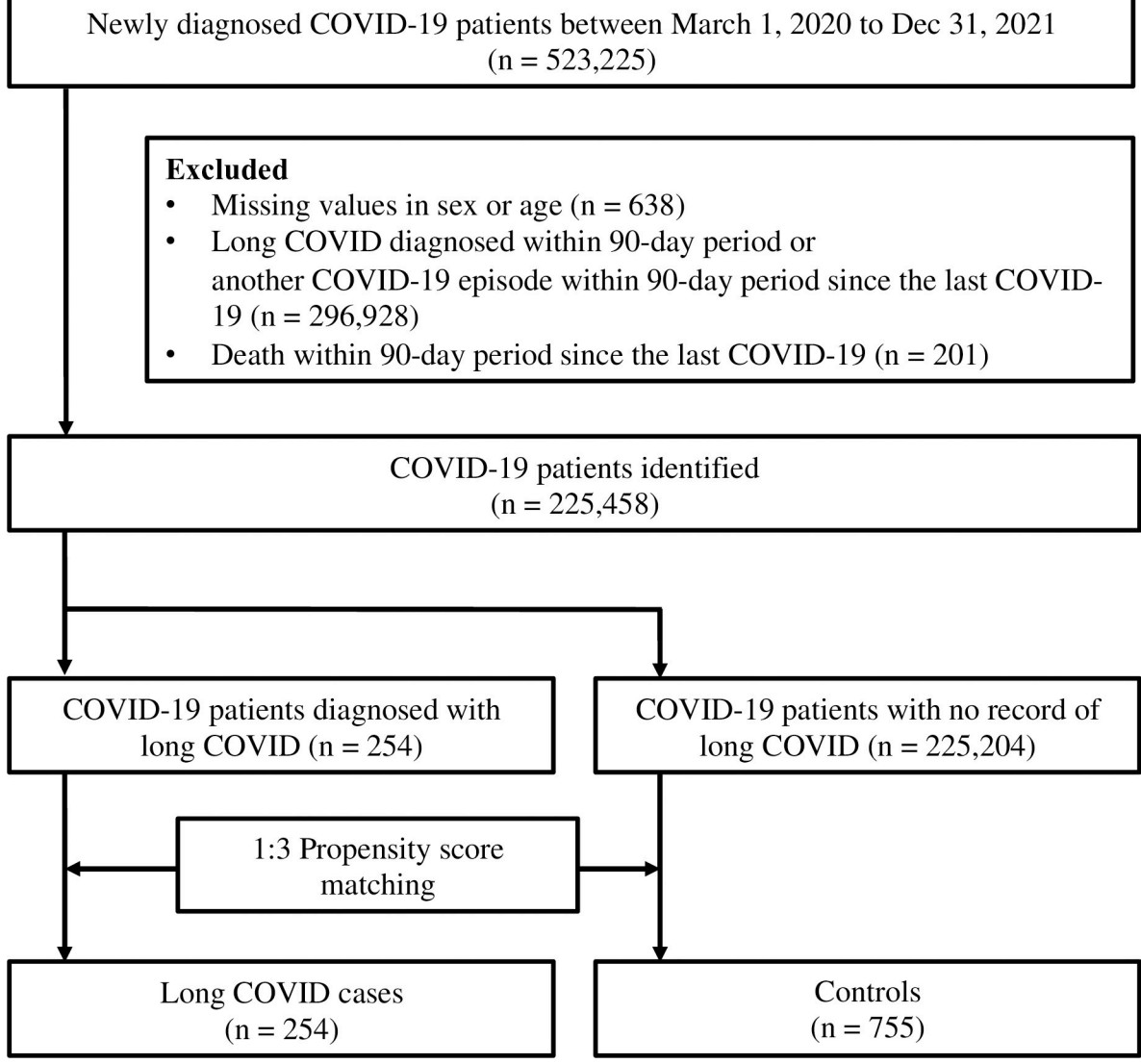

**Fig 2. Study flowchart.** COVID, coronavirus disease; COVID-19, coronavirus disease 2019.

**Table 2. Univariate and multivariate logistic models of medication exposure in patients with long COVID versus controls.**

| | After Propensity Score Matching | | | | |
| --- | --- | --- | --- | --- | --- |
| | Model 1 (Unadjusted) | Model 2 | Model 3 | Model 4 | Model 5 |
| | OR (95% CI) | OR (95% CI) | OR (95% CI) | OR (95% CI) | OR (95% CI) |
| **NSAIDs (ref. non-users)** | | | | | |
| Acute exposure[1] | 1.759 (0.996–3.107) | 1.765 (0.996–3.131) | 1.778 (1.000–3.159)* | 1.788 (1.004–3.185)* | 1.699 (0.934–3.092) |
| Chronic exposure[2] | 0.598 (0.439–0.816)** | 0.588 (0.429–0.806)** | 0.588 (0.429–0.807)** | 0.586 (0.427–0.805)** | 0.538 (0.384–0.755)*** |
| **APAP (ref. non-users)** | | | | | |
| Acute exposure[1] | 1.128 (0.774–1.643) | 1.125 (0.769–1.648) | 1.133 (0.768–1.671) | 1.136 (0.766–1.684) | 1.120 (0.748–1.678) |
| Chronic exposure[2] | 0.658 (0.464–0.932)* | 0.651 (0.458–0.926)* | 0.651 (0.458–0.925)* | 0.650 (0.457–0.925)* | 0.592 (0.409–0.857)** |
| **Antiviral drugs (ref. non-users)** | | | | | |
| Acute exposure[1] | 3.582 (1.681–7.636)** | 3.693 (1.708–7.985)*** | 3.723 (1.719–8.063)*** | 3.737 (1.723–8.106)*** | 3.749 (1.658–8.477)** |
| Chronic exposure[2] | - | - | - | - | - |

Model 1: univariate; Model 2: adjusted for sex and age group; Model 3: adjusted for sex, age group, and region; Model 4: adjusted for sex, age group, region, and CCI;

Model 5: adjusted for sex, age group, region, CCI, and underlying comorbidities.

[1] Day of COVID-19 diagnosis to 14 days later

[2] 15 days since COVID-19 diagnosis to end of follow-up (diagnosis of long COVID, diagnosis of other COVID-19 episodes, or end of observation period)

*$p < 0.05$

**$p < 0.01$

***$p < 0.001$

CI, confidence interval; APAP, acetaminophen or paracetamol; NSAIDs, nonsteroidal anti-inflammatory drugs; OR, odds ratio

### Potential relationship between NSAID use and diagnosis of long COVID

Our analysis revealed a possible link between NSAID use during acute-phase COVID-19 (0–14 days since diagnosis) and an increased risk of long COVID development. In the PS-matched cohort, individuals with long COVID were more likely to have been exposed to NSAIDs within the first 14 days after their COVID-19 diagnosis (adjusted OR [aOR], 1.79; 95% CI, 1.0–3.19; **Table 2**). Interestingly, this trend changed during chronic-phase COVID-19 (15 d post-diagnosis). In this timeframe, patients with long COVID were less likely than controls to have used NSAIDs (aOR, 0.54; 95% CI, 0.39–0.76).

### Relationship between acetaminophen or antiviral drug use and diagnosis of long COVID

Further analyses should explore the use of APAP and other antiviral drugs. We found no significant difference in APAP use during the acute phase between long COVID cases and controls (aOR, 1.12; 95% CI, 0.75–1.68). However, APAP use during the chronic period was lower in long COVID cases versus controls (aOR, 0.59; 95% CI, 0.41–0.86). Patients with long COVID were more likely to be prescribed antivirals during the acute phase (aOR, 3.75; 95% CI, 1.66–8.48). As expected, no patients with long COVID received antiviral prescriptions during the chronic period.

### Sensitivity analysis

When we included the duration of COVID-19 self-quarantine as a covariate in the analysis, the initially observed association between higher NSAID use in the acute phase and long COVID became statistically non-significant (aOR, 1.64; 95% CI, 0.90–2.99; **S2 Table**). However, the lower odds of NSAID use during the chronic period (aOR, 0.54; 95% CI, 0.39–0.76)

and lower odds of APAP use during the chronic period in patients with long COVID versus controls remained statistically significant (aOR, 0.60; 95% CI, 0.41–0.86). Additionally, the association of higher antiviral use in the acute phase with long COVID remained significant (aOR, 3.63; 95% CI, 1.60–8.25).

While a trend toward higher NSAID use in the acute phase for long COVID cases was observed, it was not statistically significant (aOR, 2.81; 95% CI, 0.62–12.77; **S3 Table**). Similarly, no significant difference was found in the relative use of NSAIDs and APAP between cases and controls during the chronic period (aOR, 1.02; 95% CI, 0.53–1.96).

## Discussion

This nationwide cohort study found that NSAID use during the acute phase of COVID-19 increased the odds of long COVID development. Patients who took NSAIDs in the first 14 days after diagnosis were more likely to develop long COVID than those who did not. This association was initially significant but became non-significant upon consideration of the duration of self-quarantine, suggesting the potential influence of disease severity, NSAID use, and long COVID risk. These findings highlight the need for the cautious consideration of NSAID prescriptions during acute-phase COVID-19.

Interestingly, this study revealed a contrasting trend in chronic NSAID use. NSAID and APAP use during chronic-phase COVID-19 were associated with a significantly lower risk of long COVID. Patients with long COVID were less likely to have used these medications than controls throughout the chronic period. This finding remained robust even after accounting for comorbidities and self-quarantine duration. These results suggest that the sustained use of these medications during the chronic phase of COVID-19 may be protective against long COVID.

Studies have suggested that NSAID use may dampen the immune response to infection [10, 11], potentially leading to inadequate viral clearance and prolonged inflammation during acute-phase COVID-19, which could contribute to prolonged COVID. This may be because NSAIDs reduce the production of immune signaling molecules and antibodies. However, during chronic-phase COVID-19, sustained NSAID use may mitigate chronic inflammation, a potential contributor to long COVID symptoms. Furthermore, the mechanisms underlying acute-phase COVID-19 involve viral replication and an acute inflammatory response. In contrast, long COVID may be driven by other factors, such as immune dysregulation [12] and chronic inflammation [13]. Although NSAID use in the acute phase of COVID-19 may suppress the body's defense mechanisms against the virus, the same medication could help modulate chronic inflammation, thereby reducing the risk of long-term symptoms.

Notably, this paradoxical effect was not observed with the use of APAP or antiviral drugs. A previous study found that APAP, an analgesic without known anti-inflammatory properties, did not suppress patients' immune responses [11]. Furthermore, our study suggests a stronger association between antiviral drug use and long COVID risk. In contrast, the association between NSAID use and the risk of long COVID was no longer significant when the COVID-19 quarantine duration was included as a covariate, although duration alone was not significantly associated with long COVID. This suggests that antiviral drugs might directly influence long COVID risk, while the relationship with NSAID use is more complex and potentially influenced by factors related to severity and initial infection management. Further studies are required to fully elucidate the dynamics and long-term effects of these drugs.

Empirical evidence suggests that factors such as female sex, old age, severe acute infection, ethnic minority, and socioeconomic status are associated with long-term symptoms after COVID-19 infection [14]. NSAID use during acute-phase COVID-19 and severe clinical

outcomes such as hospitalization with invasive ventilation, hospital mortality, acute kidney injury, and extracorporeal membrane oxygenation (ECMO) have been debated [15]. Traditionally, NSAIDs have been used in acute infections, but they may mask fever, delay diagnosis, and suppress the immune response [16]. Moreover, experimental studies have suggested that NSAIDs suppress the expression of antibodies and proinflammatory cytokines, which negatively affect the immune system and worsen the prognosis during the acute phase of infection [11]. Theoretically, NSAID use during acute-phase COVID-19 may promote the expression of ACE2, which facilitates the entry of SARS-CoV-2 into cells, thereby increasing the risk of multiple organ damage [4]. However, contrary to these concerns in the early stages of the COVID-19 pandemic, studies have suggested that acute-phase NSAID use is not associated with an increased risk of severe COVID-19 [17] but rather prevents progression to more severe disease [18]. Our study adds another layer of complexity: a possible link between acute-phase NSAID use and higher risk of long COVID. This highlights the need for further research to understand how NSAID use in acute-phase COVID-19 contributes to progression and development.

This study has strengths and limitations. First, we utilized a South Korean database encompassing extensive information about patients with COVID-19. Analyzing data from all diagnosed cases during the study period enhanced the credibility of the research. Additionally, this study employed PS matching to account for known risk factors of long COVID and minimize the influence of confounding variables on the association between NSAID use and long COVID. This study also had notable limitations. First, it included only data from October 2020 to December 2021 due to limited data availability. Second, this relatively short timeframe may not capture the impact of the emerging COVID-19 variants. The current database lacks information on COVID-19 variants, thereby preventing an impact assessment. In this study, matching by year and month of diagnosis was performed to minimize potential bias due to variants. Third, there are issues related to the diagnosis of patients with long COVID-19 who experience long-term symptoms; specifically, only a portion of them receive a formal diagnosis. This study defined long COVID cases as those diagnosed with code UOD at 3 months post-infection, potentially excluding a substantial number of actual cases [3]. Finally, the results should be interpreted carefully due to the inherent limitations of the study design. This study did not consider ICU admission or ECMO use for severe acute symptoms. Since these factors may be correlated with long COVID onset, this is a limitation. Regarding comorbidities, a slight difference was noted between cases and controls in terms of anemia (SMD = 0.20); however, the weak association between anemia and the use of painkillers and balance in other diseases that may directly be associated with painkiller prescriptions allowed us to rule out the potential for unmeasured confounders. Additionally, as mentioned earlier, limited access to medical care for long COVID symptoms restricted the available sample size, thereby impacting the analytical power. Moreover, the observed association between antiviral use and long COVID risk could be due to factors beyond the scope of this study, such as the severity of acute infection. Finally, the study did not consider over-the-counter medications such as NSAIDs and APAP, which are commonly used during both acute- and chronic-phase COVID-19. These points emphasize the need for a cautious interpretation of the results.

## Conclusions

This study identified a potential link between NSAID use during acute-phase COVID-19 and an increased risk of long COVID. However, the association between chronic-phase NSAID use and long COVID was negative, suggesting a possible protective effect. Future research is crucial to further explore the effect of NSAID use during COVID-19 in the incidence of long COVID. Investigating the biological mechanisms underlying the contrasting effects of NSAIDs

in acute- and chronic-phase COVID-19 by considering the influence of COVID-19 variants and incorporating data on medications is an important avenue for future research.

## Supporting information

**S1 Table. Study information.** AIDS, acute immunodeficiency syndrome; COVID-19, coronavirus disease-2019; HIV, human immunodeficiency virus; NSAIDs, nonsteroidal anti-inflammatory drugs.
(DOCX)

**S2 Table. Univariate and multivariate logistic models of medication exposure between long COVID cases and controls with duration of COVID as a covariate.** Model 1: univariate; Model 2: adjusted for COVID duration; Model 3: adjusted for duration, sex, and age group; Model 4: adjusted for COVID duration, sex, age group, and region; Model 5: adjusted for duration, sex, age group, region, and CCI; Model 6: adjusted for COVID duration, sex, age group, region, CCI, and underlying comorbidities. [1]Day of COVID diagnosis to 14 days later. [2]15 days since COVID diagnosis to end of follow-up (diagnosis of long COVID, diagnosis of other COVID episodes, or end of observation period). *$p < 0.05$; **$p < 0.01$; ***$p < 0.001$. APAP, acetaminophen or paracetamol; CCI, Charlson Comorbidity Index; CI, confidence interval; NSAIDs, nonsteroidal anti-inflammatory drugs; OR, odds ratio.
(DOCX)

**S3 Table. Univariate and multivariate logistic models comparing single NSAID users with acetaminophen or paracetamol (APAP) users and combined nonsteroidal anti-inflammatory drugs + APAP with APAP users.** Model 1: univariate; Model 2: adjusted for sex and age group; Model 3: adjusted for sex, age group, and region; Model 4: adjusted for sex, age group, region, and CCI; and Model 5: adjusted for sex, age group, region, CCI, and underlying comorbidities. [1]Day of COVID diagnosis to 14 days later. [2]15 days since COVID diagnosis to end of follow-up (diagnosis of long COVID, diagnosis of other COVID episodes, or end of observation period). *$p < 0.05$; **$p < 0.01$; ***$p < 0.001$. APAP, acetaminophen or paracetamol; CI, confidence interval; NSAIDs, nonsteroidal anti-inflammatory drugs; OR, odds ratio.
(DOCX)

**S1 Data.**
(XLSX)

## Author Contributions

**Conceptualization:** Tae-Hun Kim.

**Formal analysis:** Ye-Seul Lee, Heejun Kim.

**Funding acquisition:** Sunoh Kwon.

**Methodology:** Ye-Seul Lee.

**Supervision:** Tae-Hun Kim.

**Validation:** Sunoh Kwon.

**Writing – original draft:** Ye-Seul Lee, Tae-Hun Kim.

**Writing – review & editing:** Sunoh Kwon.

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
