## [Decision Letter · Decision Letter 0]

26 Aug 2024

PONE-D-24-33236Association between long COVID and acute usage of nonsteroidal anti-inflammatory drugs (NSAIDs) in COVID-19 patients: a nationwide cohort study from the National Health Insurance Service in KoreaPLOS ONE

Dear Dr. Kim,

Thank you for submitting your manuscript to PLOS ONE. After careful consideration, we feel that it has merit but does not fully meet PLOS ONE’s publication criteria as it currently stands. Therefore, we invite you to submit a revised version of the manuscript that addresses the points raised during the review process.

We look forward to receiving your revised manuscript.

Kind regards,

Seth Kwabena Amponsah, PhD

Academic Editor

PLOS ONE

Journal Requirements:

   "This study was supported by the Korea Institute of Oriental Medicine [grant number KSN2121220]."

    "This study was supported by the Korea Institute of Oriental Medicine [grant number KSN2121220]."

   "This study was supported by the Korea Institute of Oriental Medicine [grant number KSN2121220]."

4. In the online submission form, you indicated that "The data is available upon request and review by the National Health Insurance Service of Korea."

5. Please ensure that you refer to Figure 2 in your text as, if accepted, production will need this reference to link the reader to the figure.

6. Please remove your figures from within your manuscript file, leaving only the individual TIFF/EPS image files, uploaded separately. These will be automatically included in the reviewers’ PDF.

Additional Editor Comments :

Authors are requested to address all comments from reviewers.

Reviewers' comments:

Reviewer's Responses to Questions

**Comments to the Author**

1. Is the manuscript technically sound, and do the data support the conclusions?

Reviewer #1: Yes

Reviewer #2: Partly

2. Has the statistical analysis been performed appropriately and rigorously? 

Reviewer #1: Yes

Reviewer #2: Yes

3. Have the authors made all data underlying the findings in their manuscript fully available?

Reviewer #1: Yes

Reviewer #2: Yes

4. Is the manuscript presented in an intelligible fashion and written in standard English?

Reviewer #1: Yes

Reviewer #2: Yes

5. Review Comments to the Author

Reviewer #1: This is a very interesting paper.

Introduction: Concise and well written.

Methods: It was well written

To improve this manuscript, there should be some recommendations for policy and practice.

References: Referencing style should be consistent.

Reviewer #2: Concerns

1. The manuscript will benefit from a grammar check.

2. Rephrase your conclusion in the abstract. Should be straightforward….eg., Both NSAIDS and APAP are robustly associated with reduced risk of Long-term COVID-19…..

3. How and where were the controls recruited? Why did the controls have more medical conditions (covariates) than the cases? Could this have affected the outcomes observed in model 5 of Table 2? Could this potential protective effect be due to chance?

4. Your recommendations should be more specific and not generalized. Which specific molecular or biochemical aspects are required to expand on observed relationships?

6. PLOS authors have the option to publish the peer review history of their article (what does this mean?). If published, this will include your full peer review and any attached files.

Reviewer #1: No

Reviewer #2: No

---

## [Author Response · Author response to Decision Letter 0]

18 Sep 2024

Answer to the editor and reviewers’ comments

Editor’s comment

Thank you for your comments. We did this.

Thank you for your comments. We did this.

Thank you for your comments. We did this.

We didn’t change to our financial disclosure.

Thank you for your comments.

We followed the Plos one’s recommendation for formatting.

 "This study was supported by the Korea Institute of Oriental Medicine [grant number KSN2121220]."

 "This study was supported by the Korea Institute of Oriental Medicine [grant number KSN2121220]."

We don’t want to change funding information. We removed the description on fuding at acknowledgment section. We added it in the online submission system.

 "This study was supported by the Korea Institute of Oriental Medicine [grant number KSN2121220]."

Funders of this study did noting so we added following statement in the submission system.

4. In the online submission form, you indicated that "The data is available upon request and review by the National Health Insurance Service of Korea."

The database data from the National Health Insurance Service undergoes a special procedure for anonymization and can only be accessed at a designated location for a limited time period. As of September 2024, the access rights to the data we requested have already been revoked, making it inaccessible. Considering that health insurance data contains sensitive personal information and access is not always permitted to protect privacy, we have described it as follows:

The data included in this study from the National Health Insurance Service in Korea can only be accessed at a designated location for a limited time period for protection of personal information so it is available upon request and review by the National Health Insurance Service of Korea.

5. Please ensure that you refer to Figure 2 in your text as, if accepted, production will need this reference to link the reader to the figure.

We added it in the manuscript.

6. Please remove your figures from within your manuscript file, leaving only the individual TIFF/EPS image files, uploaded separately. These will be automatically included in the reviewers’ PDF.

We removed them.

We added captions for Supporting information files as follows:

Supporting Information files

Supplementary Table 1. Study information

COVID, coronavirus disease-2019; NSAIDs, non-steroidal anti-inflammatory drugs; AIDS, acute immunodeficiency syndrome; HIV, human immunodeficiency virus.

Supplementary Table 2. Univariate and multivariate logistic models of medication exposure between long COVID cases and controls with duration of COVID as a covariate

Model 1, univariate; Model 2, adjusted for COVID duration; Model 3, adjusted for COVID duration, sex, and age group; Model 4, adjusted for COVID duration, sex, age group, and region; Model 5, adjusted for duration, sex, age group, region, and CCI; Model 6, adjusted for COVID duration, sex, age group, region, CCI, and underlying comorbidities

1Day of COVID diagnosis - 14 days after COVID diagnosis

215 days since COVID diagnosis - end of follow-up (diagnosis of long COVID, diagnosis of other COVID episodes, or end of observation period

*p <0.05; **p <0.01; ***p <0.001

NSAIDs, nonsteroidal anti-inflammatory drugs; CI, confidence interval; OR, odds ratio; APAP, acetaminophen or paracetamol

Supplementary Table 3. Univariate and multivariate logistic models comparing single NSAID users with acetaminophen or paracetamol (APAP) users and combined nonsteroidal anti-inflammatory drugs + APAP with APAP users

Model 1: univariate; Model 2: adjusted for sex and age group; Model 3: adjusted for sex, age group, and region; Model 4: adjusted for sex, age group, region, and CCI; Model 5: adjusted for sex, age group, region, CCI, and underlying comorbidities

1Day of COVID diagnosis - 14 days after COVID diagnosis

215 days since COVID diagnosis - end of follow-up (diagnosis of long COVID, diagnosis of other COVID episodes, or end of observation period

*p <0.05; **p <0.01; ***p <0.001

NSAIDs, nonsteroidal anti-inflammatory drugs; CI, confidence interval; OR, odds ratio; APAP, acetaminophen or paracetamol

There is no retracted reference. Thank you for your comments. 

Reviewer Comments

Association between long COVID and acute usage of nonsteroidal anti-inflammatory 2 drugs (NSAIDs) in COVID-19 patients: a nationwide cohort study from the National 3 Health Insurance Service in Korea

Concerns

1. The manuscript will benefit from a grammar check.

Thank you for your comments. We already edited our manuscript before submission through a professional English editing service. But as you commented, we checked grammar once again.

2. Rephrase your conclusion in the abstract. Should be straightforward….eg., Both NSAIDS and APAP are robustly associated with reduced risk of Long-term COVID-19…..

Dear reviwer,

We added this point based on your comments as follows:

Conclusion: While this study suggests a possible links between NSAID use in acute infection stage and higher risk of long COVID as well as both NSAIDs and acetaminophen uses during the chronic period and lower risk of long COVID, the association was not statistically significant.

3. How and where were the controls recruited? Why did the controls have more medical conditions (covariates) than the cases? Could this have affected the outcomes observed in model 5 of Table 2? Could this potential protective effect be due to chance? 

Dear reviewer, 

Thank you for your valuable comments. First of all, the control group was not radomly selected; rather, it was selected through propensity score matching from the same database and within the same time period as the observational group. These individuals had COVID-19 but did not take NSAIDs, and they were chosen in a way that controlled for other confounding factors to ensure comparibility between the two groups. 

The mathing method allowed three controls per patient, matching for the exact age group, sex, CCI score, and the year and month of COVID-19 episode. For the underlying disease severity, we used the CCI score as an exact condition; this allows for a variation in the type of the underlying disease, which in our study encompassed a wide range of diseases, while matching the overall severity.

The result showed that there was no difference between the two groups except for anemia (SMD = 0.2). It is true that anemia may cause symptoms which may overlap with chronic fatigue, and the cases show higher morbidity of anemia compared to controls. However, other comorbidities which may cause more severe outcomes were balanced between the two groups, which helps us rule out the potential of confounding due to comorbidities. Furthermore, there is no direct relationship between anemia and use of painkillers, which rules out the potential unmeasured confounding between the baseline characteristics and interventions.

Based on the reviewer’s comment, we revised the manuscript as follows:

After applying PS matching to create comparable groups, a balance was achieved for demographic factors such as sex and age group, CCI, underlying comorbid conditions except for anemia, which the cases showed higher comorbidity than controls, and the year and month of the COVID-19 episode (Table 1). 

While there was a slight difference between cases and controls in the comorbidity of anemia (SMD = 0.20), the weak association between anemia and the use of painkillers and balance in the other diseases which may directly be associated with painkiller prescriptions, allowed us to rule out the potential of unmeasured confounding.

These individuals did not have more medical conditions, as can be confirmed from Table 1, which shows no differences between the two groups in terms of each medical condition after propensity score matching. When propensity score matching is performed, differences related to those factors between the groups disappear, which means they naturally do not affect the results. Therefore, it is difficult to consider the “potential protective effect” you mentioned as a mere coincidence

4. Your recommendations should be more specific and not generalized. Which specific molecular or biochemical aspects are required to expand on observed relationships?

Dear Reviewer, thank you for your comments. In fact, the exact mechanism of how long COVID develops is still not fully understood, and it is also challenging to propose a clear mechanism for how NSAIDs might worsen the symptoms of acute COVID-19 infection while having a protective effect against long COVID with prolonged use. Since this study is based on patient data, it would be difficult to specify a particular molecular or biochemical aspect for future research based solely on our results. However, to avoid any confusion in expression, we would like to clarify that section by rephrasing it as follows. We hope this meets your expectations.

Future research is crucial to further explore the effect of NSAID in the acute stage of COVID-19 infection and its molecular mechanism on the incidence of long COVID.

---

## [Decision Letter · Decision Letter 1]

29 Sep 2024

PONE-D-24-33236R1Association between long COVID and acute usage of nonsteroidal anti-inflammatory drugs (NSAIDs) in COVID-19 patients: a nationwide cohort study from the National Health Insurance Service in KoreaPLOS ONE

Dear Dr. Kim,

Thank you for submitting your manuscript to PLOS ONE. After careful consideration, we feel that it has merit but does not fully meet PLOS ONE’s publication criteria as it currently stands. Therefore, we invite you to submit a revised version of the manuscript that addresses the points raised during the review process.

We look forward to receiving your revised manuscript.

Kind regards,

Seth Kwabena Amponsah, PhD

Academic Editor

PLOS ONE

Journal Requirements:

**Additional Editor Comments:**

Whiles the manuscript has improved significantly in terms of scientific content, the entire manuscript will still benefit from English-language editing. The Journal provides such services.

I use the Abstract as example; "While this study suggests a possible links between NSAID use in acute infection stage and higher risk of long COVID as well as both NSAIDs and acetaminophen uses during

the chronic period and lower risk of long COVID,......"

Reviewers' comments:

Reviewer's Responses to Questions

**Comments to the Author**

1. If the authors have adequately addressed your comments raised in a previous round of review and you feel that this manuscript is now acceptable for publication, you may indicate that here to bypass the “Comments to the Author” section, enter your conflict of interest statement in the “Confidential to Editor” section, and submit your "Accept" recommendation.

Reviewer #1: All comments have been addressed

Reviewer #2: All comments have been addressed

2. Is the manuscript technically sound, and do the data support the conclusions?

Reviewer #1: Yes

Reviewer #2: Yes

3. Has the statistical analysis been performed appropriately and rigorously? 

Reviewer #1: Yes

Reviewer #2: Yes

4. Have the authors made all data underlying the findings in their manuscript fully available?

Reviewer #1: Yes

Reviewer #2: Yes

5. Is the manuscript presented in an intelligible fashion and written in standard English?

Reviewer #1: Yes

Reviewer #2: Yes

6. Review Comments to the Author

Reviewer #1: (No Response)

Reviewer #2: (No Response)

7. PLOS authors have the option to publish the peer review history of their article (what does this mean?). If published, this will include your full peer review and any attached files.

Reviewer #1: No

Reviewer #2: No

---

## [Author Response · Author response to Decision Letter 1]

8 Oct 2024

Reply to the editorial comment

Whiles the manuscript has improved significantly in terms of scientific content, the entire manuscript will still benefit from English-language editing. The Journal provides such services.

I use the Abstract as example; "While this study suggests a possible links between NSAID use in acute infection stage and higher risk of long COVID as well as both NSAIDs and acetaminophen uses during

the chronic period and lower risk of long COVID,......"

Dear editor

Thank you for your comments. We edited our manuscript in a professional editing service and you can check the certificate by the company in the attached file.

We revise the abstract section as you suggested:

Although this study suggests a possible link between NSAID use in the acute COVID-19 infection stage and a higher risk of long COVID as well as both NSAID and acetaminophen use during the chronic COVID-19 period and a lower risk of long COVID, the association was not statistically significant.

---

## [Editor Report · Decision Letter 2]

9 Oct 2024

Association between long COVID and nonsteroidal anti-inflammatory drug use by patients with acute-phase COVID-19: A nationwide Korea National Health Insurance Service cohort study

PONE-D-24-33236R2

Dear Dr. Kim,

We’re pleased to inform you that your manuscript has been judged scientifically suitable for publication and will be formally accepted for publication once it meets all outstanding technical requirements.

Kind regards,

Seth Kwabena Amponsah, PhD

Academic Editor

PLOS ONE
---

## [Editor Report · Acceptance letter]

23 Oct 2024

PONE-D-24-33236R2 

PLOS ONE

Dear Dr. Kim, 

I'm pleased to inform you that your manuscript has been deemed suitable for publication in PLOS ONE. Congratulations! Your manuscript is now being handed over to our production team.

Kind regards, 

on behalf of

Prof. Seth Kwabena Amponsah 

Academic Editor

PLOS ONE